# Mercury Exposure among E-Waste Recycling Workers in Colombia: Perceptions of Safety, Risk, and Access to Health Information

**DOI:** 10.3390/ijerph18179295

**Published:** 2021-09-03

**Authors:** Maria Jensen, David Andrés Combariza Bayona, Kam Sripada

**Affiliations:** 1Department of Public Health and Nursing, NTNU—Norwegian University of Science and Technology, 7491 Trondheim, Norway; jens1802@umn.edu; 2Departamento de Toxicología, Facultad de Medicina, Universidad Nacional de Colombia, Bogotá 11321, Colombia; 3Centre for Global Health Inequalities Research (CHAIN), NTNU—Norwegian University of Science and Technology, 7491 Trondheim, Norway

**Keywords:** occupational health, global health inequalities, social determinants of health, access to health information, heavy metal, lightbulb, WEEE, RAEE, mercurio, salud ocupacional

## Abstract

Exposures to the toxic element mercury (Hg) are exceptionally high among recycling workers globally. Recycling is a growing sector in Colombia, yet workers who directly handle e-waste are often unaware of the risks of exposure to mercury from post-consumer lighting products (e.g., fluorescent lamps). This qualitative study aimed to understand how recycling workers perceive their own risks from mercury exposure and how they find information about these risks, through interviews (*n* = 35) at the three largest formal recycling facilities in Colombia. Workers’ risk perception was generally disconnected from their likely actual exposure to mercury, instead often seen juxtaposed to co-workers who worked more directly with hazardous waste. Recycling workers, who were predominantly men from lower-income socioeconomic backgrounds, had limited knowledge of health risks due to mercury exposure and were more likely to receive health-related information from informal sources. Over a third of interviewees had searched online for information about occupational health risks of mercury, but these searches were perceived as unsatisfactory due to information being difficult to find, not available in Spanish, or related to mercury exposure via seafood or mining rather than recycling. Workers expressed (over)confidence in personal protective equipment and concern about frequent employee turnover. This study points to weaknesses in environmental health literacy and public health communication around toxic exposures to mercury in the workplace. Stronger regulation and enforcement are needed to prevent toxic exposures and promote worker health equity.

## 1. Background

Recycling workers have exceptionally high exposures to the toxic element mercury (Hg) globally [1,2,3,4,5,6,7,8,9,10]. Yet there is no known threshold for safe mercury exposure [1] and evidence indicates that current thresholds for tolerable workplace exposure may be inadequate to protect workers [2,5]. Globalization has increased the transfer of hazardous products without a corresponding strengthening in safeguards for their proper disposal; e-waste (i.e., discarded electric and electronic items) is one of the clearest examples of this phenomenon, leading to unequal health risks in the Global South [11,12].

Metallic or elemental mercury is an essential component of many products, including fluorescent lamps, thermometers, some auto parts, electrolytic cathodes, and batteries, all of which are commonly recycled [13,14,15]. When fluorescent light bulbs are recycled, which includes breaking the bulbs into small shards, elemental mercury vapors are released at a high rate and can linger in the atmosphere for weeks [3,7,9,16,17]. Without adequate safety measures, the emitted mercury has the potential to contaminate the environment near workers who handle the waste [15,18,19,20]. Elemental and inorganic mercury, most common in occupational exposure, can be absorbed into the human body through inhalation, digestion, or dermal exposure [13]. In the case of recycling, routes of mercury exposure include inhalation of vapors that are emitted during the recycling process or skin contact with broken materials [16,21,22]. Mercury is a non-essential mineral toxic to all organs in the human body, primarily affecting the central nervous system and the kidneys [14,23,24] and with symptoms observed at both low-level chronic exposures [22,25,26] and high exposure [22].

In Colombia, recycling is a growing sector [27,28], but data on exposure to mercury from e-waste including post-consumer lighting products is lacking. Indeed, the topic of toxic occupational exposures is sensitive, and therefore research on it in Colombia has been limited. Since the government began to phase out inefficient incandescent bulbs in 2007, the use, production, and ultimate disposal of fluorescent light material has increased substantially [29,30,31]. Since 2013, national laws in Colombia have sought to recognize and reduce mercury-related health risks, including by ending the import and export of most products that contain mercury, such as light bulbs, thermometers, sphygmomanometers, barometers, and batteries [32,33,34,35]. Colombia ratified the Minamata Convention in 2018, and the United Nations has also made major investments to reduce the use of mercury in Colombia [36,37,38,39]. But despite regulation efforts, mercury contamination persists as a public health threat, with high levels of exposure in workers [28,40,41,42] and a lack of consistent and enforced safety regulations [43,44].

Perception of risk within the lay population often has little relation to their objective risk [45]. Risk perception research seeks to better understand a community’s perception of risks, which can help in designing risk communications and understanding what information workers may lack in order to protect themselves [46,47,48]. Such studies are especially relevant in occupational settings in which risk perceptions affect workers’ safety behavior, and thus likely influence their actual risk [45,49].

Little is known about how individuals exposed to mercury in the workplace come to perceive risks related to mercury exposure via recycling [1,2,3,4,5], particularly in South America [50,51,52,53]. In the workplace, misperceptions of risk may result in overconfidence in one’s own safety, inappropriate safety behavior, or potentially unnecessary stress about occupational risks [45,54,55]. Research on perceptions of mercury-related health risks has primarily been conducted among populations with fish-based diets, indigenous communities, pregnant women, those who are exposed to dental amalgams, gold miners, and people in villages near artisanal gold mines [56,57,58,59,60,61,62]. Among these populations, communication has been shown to play a key role in the perception of risk [63,64,65,66].

This is one of the first studies to investigate how recycling workers in two Colombian cities who handle mercury-containing post-consumer lighting materials perceive their own risks from mercury contamination in the workplace, and how they find information about this risk.

## 2. Methods

### 2.1. Participants

Interviews were conducted on site at one of three recycling centers. All interviewees (*n* = 35) were adults (18 years or older) who met the inclusion criterion of currently holding a job in one of these companies that involves working directly with or in close proximity to mercury-containing waste materials, based on a purposive sampling approach [67]. All employees working directly with recycling the lighting material (“operators”) who were on site during the days of data collection were interviewed. In addition, interviews were conducted with “non-operators,” such as supervisors and administrative staff who work near mercury-containing hazardous material, truck drivers who handle the material, transportation coordinators, safety coordinators, security officers, engineers, and workers who recycle other materials but are exposed to the environment of the lightbulb treating process. Participation rate was 100%.

Participants self-reported their neighborhood’s economic stratum, assigned for tax purposes, ranging from 1 to 5, with 1 indicating the lowest-income neighborhoods. This information was used as a proxy for socioeconomic status.

### 2.2. Risk Perception Theory

In this study, risk is defined as the probability of an outcome due to a particular exposure [68]. Risk perception is assessed using four components in the chain of risk perception that affect risk behavior [69]:

*Severity—*How severe do they believe the threat to be?

*Susceptibility—*How at risk do they believe they personally are to the threat?

*Response efficacy—*Do they trust that the suggested intervention is effective?

*Self-efficacy—*Do they believe that they are able to perform recommended actions to protect themselves?

### 2.3. Study Setting

The research was based in Bogotá and Cali, Colombia, at three recycling facilities that together process 72.6% of all lighting materials recycled in Colombia [70]. Names of the three companies have been anonymized for this study. Interviews were conducted between December 2019 and February 2020, twice at company 1, and once each at Companies 2 and 3.

### 2.4. Data Collection

A total of *n* = 35 semi-structured interviews were conducted in Spanish. Two interviewers were present for each interview. Interviews concluded when the interviewers felt they had reached saturation of information for that participant. Interviews length ranged from 6 to 30 min, with an average of 13 min. Each interview was audio recorded, transcribed in Spanish, and reviewed by a native Spanish speaker. All interviews were included in the analysis. Approximately 10% of 3 interviews from company 1 could not be transcribed due to poor audio quality; however, no key data is believed to have been lost.

### 2.5. Data Analysis

All data was analyzed after each interview round using a grounded theory approach [71]. NVivo (QSR International, Melbourne, Australia) was used to analyze the data, using a bottom-up approach [72]. Text was coded for themes and subthemes, assessed for word frequencies, and cross-tabs analyzed for relationships with demographic data, specifically gender, length of employment, age, and self-reported economic strata. Audio recordings were revisited as necessary, for example for determining risk perception scores.

## 3. Results

### 3.1. Demographic Information

A total of *n* = 35 interviews were conducted across three recycling centers, of which *n* = 21 were with operators who manually handle lighting recycling material. Interviewees were most commonly between 30 and 39 years old and had worked at the company between 3 days and 14 years, with the majority having spent under one year with the company. Most operators had completed a high school education. Interviewees were mostly from the tax strata 2 or 3, and operators tended to be from lower strata neighborhoods than non-operators. Additionally, the workers from company 1 tended to be from lower strata than workers from companies 2 and 3.

The gender distribution was 22 males to 13 females. There was an inevitable gender disparity, as there were more men working in this field than women.

### 3.2. Differences by Gender, Education, Socioeconomic Status, and Age

Results from cross tabulation of demographic data (gender, education, socioeconomic status, and age) with response codes were mostly insignificant, with the exception of a few codes. There were no substantial relationships related to age or education.

Personal protective equipment was brought up 34% more frequently by females than males and general safety measures was brought up 47% more by females than males. Mentions of being worried arose 40% more frequently from interviews with females than with males.

Numerous interviewees reported a lack of awareness within the company of the risks of mercury exposure. These responses had a relationship with socioeconomic status: 67% of the strata 5 interviewees (highest-income) brought up the issue and 42% of the interviewees from strata 3—as opposed to 13% in strata 2 and 0% in strata 1.

Transportation and handling of the light bulbs as they relate to workers’ mercury exposure was mentioned by 66% of strata 5 (highest-income) interviewees, and 50% from strata 3. This result is likely related to the fact that employees in management self-reported to be from higher-income social strata.

Males mentioned specific risky and preferable behaviors happening in the workplace 33% more frequently than females did; however, this was a biased result, as the majority were generated from interviews in company 1, which was 90% male.

### 3.3. Sources of Health Information

Operators and non-operators received health information from different sources (Figure 1). Operators were more likely to get information from informal sources such as the internet, a friend, a colleague, or a family member. Non-operators were more likely to say they were trained at work or had used a formal source such as materials from the National Institute for Occupational Safety and Health (NIOSH), the World Health Organization (WHO), or Lúmina, a Colombian company that joins different companies in the lighting sector and works as a source for information on recycling lighting material.

A third of all interviewees had used the internet to supplement the information they received at work. Of those who searched online, none were satisfied with what they found. The security coordinators from each company reported believing that the information was not easily accessible online for the operators because the information was not available in Spanish, the workers did not know how to search properly for such information, or the information was too difficult to digest for the education level of the operators. The operators who searched online either said they were unable to find the information they wanted to find, all the information they found was either related to exposure to mercury via fish or mining, or the information was not clear to them. Among the operators who searched the internet for risk information, three-quarters were highly unconfident in their knowledge about risks of mercury exposure in general, unsure of their personal risk, and/or expressed fear.

### 3.4. Risk Perception

Results are provided for each of the four components of risk perception: severity, susceptibility, response efficacy, and self-efficacy. Responses were scored from 1 to 5, with 5 being most concerned.

#### 3.4.1. Severity

The average severity scores for each company were as follows: company 1: 3.5, company 2: 4.9, and company 3: 4.3. Company 1 had a wide range of answers (range: 1 to 5), whereas the other two were more consistent (range: 3 to 5). The highest average concern was in company 2.

Severity scores ranged from 1 to 5 (most severe), as exemplified in the following excerpts regarding workplace mercury exposure: 

1: “*I don’t think it is dangerous.*”(interview 8)

2: “*It’s not a 100% risk.*”(interview 1)

3: “*It’s a bit dangerous to handle.*”(interview 14)

4: “*It’s an element that stays in the environment and the human body, in food … for this reason, it’s a high risk.*”(interview 27)

5: “*He got fucked up from mercury”; “people are really afraid of this.*”(interview 6)

#### 3.4.2. Susceptibility

A higher proportion of operators than non-operators believe they have only low levels of exposure, although the operators are much closer to the source or work directly with the lighting material. Operators also varied more in their perception of their personal exposure than non-operators. Among operators, 70% (*n* = 13) said they had a low level of exposure, whereas 14% (*n* = 3) said they had medium exposure, and 24% (*n* = 5) high exposure. Among non-operators, 50% (*n* = 6) said they had low exposure, 50% (*n* = 6) medium exposure, and 0 high exposure.

In general, workers tended to measure their level of exposure by comparing it to that of other workers who are perceived to work more closely with the lighting material. In company 1, those working on the third floor disassembling other types of e-waste consistently reported having an insignificant level of exposure due to their lack of direct contact with the lighting material, which occurred two floors below in an open facility. In company 3, it was common to hear the operators compare themselves with the only worker who operates the bulb crushing machine. Using this comparison, they reasoned that their exposure was minimal. Nearly all non-operators mentioned that their exposure was nothing or close to nothing in comparison to the operators and did not distinguish between operator roles when talking about their own level of exposure. Though they stated their exposure was low, some administrative staff mentioned having had symptoms of exposure to mercury such as a “sweet” sensation in the mouth.

“*I try to not be here during peak times*” (interview 12), said one non-operator, meaning risk avoidance when the operators are running the glass breaking machine.

#### 3.4.3. Response Efficacy

Workers from all groups expressed high confidence in their personal protective equipment (PPE). Workers reported using varying types of PPE: full-face or half-face masks with mercury-specific filters, disposable overalls, glasses, anti-cut gloves, steel-toed boots, hearing protection, and helmets. The use of protective gear varied within and between the three companies. In company 1, all operators reported using a half-face mask and overalls with a hood. In companies 2 and 3, some workers who have more contact with the lighting material reported using a full-face mask and full-body overalls with a hood.

The only doubts came from company 1, where one operator voiced a doubt that the filters were preventing the mercury from passing into their body. A non-operator worried that the masks being used in company 1 were not effective, as they did not cover the whole face, leaving bare skin exposed.

Otherwise, there was an overall sense among the interviewees that the PPE alone was sufficient to protect them from mercury exposure. Of the 35 interviewees, 27 expressed high confidence in the protective equipment, such as the following: 

“*For now, it’s safe because we use the*
*personal protective equipment.*”(Interview 1)

“*Anyway, we use the mask, gloves, everything—it’s all secure.*”(interview 10)

“*If I use the*
*personal protective equipment, I don’t think it could affect me much.*” (Interview 17)

“*Could be (dangerous)…but with all the elements of security they give us… they provide a lot of protection*.”(Interview 25)

#### 3.4.4. Self-Efficacy

Perceptions of self-efficacy—the last link in the chain of risk reduction [68]—were similar between companies 2 and 3 and between gender groups. In company 1, workers expressed concern over their limited individual ability to implement the standard safety measures, such as overheating due to PPE, excessive sweating, difficulty breathing with the mask on, allergies to the mask, and general discomfort. In company 1, removing of masks for various reasons seemed to be of higher concern, and perceived self-efficacy in their ability to use the PPE consistently appeared to be lower. Some workers were concerned about being unable to communicate while wearing filters in the warehouse, as well as workers using expired filters. One administrative worker in this company mentioned they did not have their own mask with a mercury filter for visits to the warehouse, where the recycling is performed.

### 3.5. Perceived Health Risks

Health risks related to mercury exposure were perceived differently by operators and non-operators. Non-operators responded to the question about health risks of mercury with ideas about damage to the central nervous system, skin, blood, and memory loss, as well as the concepts of accumulation in the body, lack of a cure, and delayed effects. The operators the other hand mentioned cancer most frequently, then damage to organs (lungs, skin, kidneys, liver, eyes, stomach), the brain, blood, and infertility. Overall, the respondents had much variation among them, but no substantial differences between the three companies or between genders. None of the 35 interviewees mentioned risks of mercury exposure for potential pregnancy or harm to fetuses, other than to mention the risk of male infertility.

When asked about their health status in relation to their work, some operators responded saying they felt there was no reason for concern as they had no symptoms, such as:

“*For the moment I feel fine; I don’t have any symptoms or anything, no headaches*.”(interview 31)

### 3.6. Rotation of Personnel

When asked about difficulties for safety in the companies, interviewees at all three companies reported a high rate of rotation of workers. For example, one of the workers who handles lighting material mentioned that 15 to 20 people have passed through the 4 positions in the area of light bulb recycling since they started working one month prior (interview 31). Another employee reported:

“*Some workers last three days*”; “*Others come and by mid-day they are gone.*”(interview 5)

This high rotation was said to be difficult, as it wastes the time of the security coordinators; wastes filters, as they are not reused for different workers; and means that the workers handling the lighting material have little experience:

“*Because here there is a lot of rotation of personnel, and so maybe the new ones just don’t think that the mask is important*.”(interview 11)

Several interviews provided insights as to causes of high rotation in personnel. In company 1, three interviewees expressed that their work was not a respected profession. One interviewee described how they believed operators felt their job was perceived by outsiders as similar to sorting through trash in the streets to sell recyclable material, which the interviewee assumed was a reason why workers do not stay with the company:

“*[It’s] the work; here [we process] also cardboard, plastic, glass … well, people see it as a recycling company*”; “*… like a street recycler …*”; “*They don’t see their whole life doing this*.”(Interview 5)

In relation to the perception of the work, there was a clear difference between company 1 and company 3: whereas in company 1 there was mention by operators of discontentment in the perception of their work, in company 3, four workers reported that they liked that their work was helping the environment. Fear of mercury contamination was mentioned in four of the 35 interviews as a reason for people leaving their job, primarily at company 1.

Wages were mentioned multiple times as an issue related to people quitting employment at company 2. An operator mentioned that he and his colleagues thought that since their job involved handling dangerous material, they should be paid more than minimum wage. An administrative staff member said they used to try to recruit informal recycling workers, thinking they would appreciate having a formal position with health insurance and safer work conditions, yet the result often was that they preferred to work on their own and earn about $50 USD more each month.

Hard physical work was a theme in all three companies. Some workers were not happy with the job because it was physically very demanding:

“*There are people who aren’t a good fit. In the plant, it’s a matter of handling loads. It is not a simple issue. We have many mechanical tools but you still have to move the 250–300 kg tub to get it on the scale*.”(Interview 22)

Workplace safety improvements recommended by recycling workers to reduce hazardous exposures to mercury are presented in Table 1. These recommendations expressed during interviews touch on training needs, issues related to the physical facility, client interactions, and lack of biomonitoring.

The interview excerpts provided in the Results section have been anonymized and translated into English; original quotations in Spanish are provided in Table 2.

## 4. Discussion

Interviews with employees at three major recycling companies in Colombia identified gaps in workers’ knowledge about the risks of mercury and, among those working mostly closely with mercury-containing recyclables, a disconnect between their own level of concern and their likely actual exposure. Overconfidence in PPE used on site and lack of familiarity with health risks associated with mercury exposure may foster a sense of safety among workers without regard for their likely actual exposure levels, which are untested. Still, most operators reported that they believed they had at least a low level of exposure to mercury. Constant employee turnover was cited frequently as a safety concern. While over a third of employees had searched online for information about occupational health risks, these searches were often unsatisfactory. Weaknesses in several links in the chain of risk perception—specifically in workers’ misperceptions of severity, susceptibility, and self-efficacy—point to needs for stronger public health communication for mercury exposure in the lighting and e-waste recycling industry. This can support the prevention of mercury releases related to recycling processes in Colombia, a stated policy priority [75].

Lack of data about occupational hazards in Latin America likely hides additional injuries and negative health effects [76]. Results from this study help fill this knowledge gap with relevant information on environmental health literacy and risk perception, which may prevent occupational injury. Employees directly handling mercury-containing materials had a lower awareness of the risks from mercury exposure than non-operators (e.g., administrative personnel, employees in other sections), and more of them were unconcerned. The operators who were concerned spoke of fear and feeling uninformed about their risks. The non-operators had more accurate and confident responses to the questions about health effects of mercury. Most operators believed that mercury causes cancer, though evidence for this is limited [15,77,78]. Operators often believed they were safe due to an absence of what they considered to be alarming symptoms, and lacked an understanding that the effects from chronic mercury exposure may not be noticed right away. Risks to pregnancies were not mentioned by any interviewees, although mercury exposure is hazardous during pregnancy and for women who may become pregnant [79,80,81,82]. Workplace culture and safety standards also appeared to play a role in risk perception. Perception of one’s job may itself be related to safety behavior of workers, as those who have worse opinions about their occupation report having less caution in their activities [51].

These findings can be used to create targeted and effective public health messages. Based on Cho’s *Health Communication Message Design* [69], interviews indicated weaknesses in three of the four links of the chain of risk perception. First, workers too frequently perceived that the health risks from mercury exposure at the workplace were not severe. Second, in general they perceived themselves as not being personally susceptible to the risks, often comparing themselves to a co-worker who worked more closely with hazardous waste. These misperceptions of severity and susceptibility likely contribute to an inflated sense of safety. Third, workers reported difficulty using their PPE correctly, especially in company 1, indicating a low belief in their capacity to protect themselves from mercury exposure. By contrast, workers generally perceived that the PPE provided to them, if used properly, was effective. Though confidence in PPE was high overall, the equipment and implementation may not meet safety standards. For example, several interviewees mentioned that most workers have some skin exposed, particularly in company 1, while this is not in accordance with safety standards [83].

Risk perception studies find that workers often trust informal sources over authorities on the subject, as well as underestimate severe risks and overestimate minor ones. Operators at all three companies reported relying more on informal sources, such as the internet and co-workers. Fear and uncertainty about workplace risks were commonly expressed by operators who searched for information online. By contrast, non-operators, who on average had a higher-income socioeconomic status and less staff turnover, have more opportunity to access and receive information from their employers about occupational hazards. Yet while non-operators demonstrated greater health literacy regarding mercury, it is operators who in fact have direct contact with hazardous recycling materials, highlighting occupation as a critical social determinant of health. Public health messaging about mercury exposure in the workplace, especially in easily understandable infographics (such as Figure 2, Figure 3 and Figure 4) and videos online available in Spanish are needed to reach workers in the places where they are seeking health information.

### 4.1. Policy Relevance 

The growing acknowledgement of the scale of mercury’s human health impact has spurred global health action [80,81,82]. The Minamata Convention on Mercury, adopted by the United Nations in 2013, currently has 128 signatories and requires practical steps toward the reduction of mercury which, if implemented, are expected to averted health losses at a large scale [84,85,86]. The International Labor Organization filled gaps left in the Minamata convention, such as creating a supervisory mechanism for the convention’s implementation [87]. In 2015, the European Union placed restrictions on the use of mercury in products with the Restriction of Hazardous Substances Directive 2015/863/EU and has phased out many products containing mercury such as medical and lighting devices [88,89].

As a signatory of both the Minamata and Basel Conventions, Colombia has decreased mercury imports into the country from an average of 100,000 kg per year down to 5000 kg in 2017 [90,91,92,93] and participates in Latin American policy dialogue on proper handling of hazardous post-consumer goods [92,94,95]. To date, research on mercury exposure in Colombia has primarily focused on small-scale gold mining, as it is widespread and under-regulated [96,97,98,99,100,101,102,103,104]. Occupational injuries in Latin America are likely underreported, due to a lack of comprehensive surveillance [105] and, in the case of mercury, the latency between exposure and adverse health effects. Informal e-waste recycling involves exposure to mercury alongside lead and other hazards [106,107]. As recycling of mercury-containing products and other hazardous waste continues to grow in Colombia, public health planning and policy must use both formal and informal information channels to disseminate health risk information to better inform members of the recycling community of their occupational health hazards. Language, literacy, and cultural factors may make informal communication channels easier to access and therefore valuable for communicating occupational health messages.

The World Health Organization has found significant losses in Disability Adjusted Life Years (DALYs) from decreases in IQ related to mercury exposure worldwide, linked to large global productivity losses [85,108,109,110,111,112]. Such findings highlight a major health inequality for developing countries, where exposures to mercury are on average higher, economies are weaker, and appropriate safeguards may not be in place [113]. The recycling process of florescent bulbs is expensive (costing about 0.05 to 0.10 USD per bulb), which makes this business especially challenging in developing economies where making a profitable margin is more difficult [114,115]. Whereas in some European countries the price of the recycling process is included in the sale price of the product, in Colombia this is not the case, and companies must manage to make a profit solely by selling the recycled material [116,117]. A low margin for the recycling companies makes implementing extensive safety procedures more challenging in countries with developing economies like that of Colombia [114,118]. This is true for mercury, as well as occupational hazards more broadly in low- and middle-income countries [119].

### 4.2. Strengths and Limitations 

During interviews, employees were asked to share perceptions of health risks which may have been at odds with their wish to feel safe at work or which they may prefer remain undisclosed to their employers. Interviews in general were conducted in a private room (with the exception of one interview) but subject to loud noises during working hours. Though interviews were conducted in private and the interviewees were informed their responses were not to be shared, there could have been biases due to the location and time of the interviews (during the work day, at the place of employment). Two interviewers were present for each interview, of whom one was fluent in Spanish but not Colombian Spanish, together with an interviewer from the Universidad Nacional de Colombia. All transcriptions were reviewed by Colombian native Spanish speakers.

The study is strengthened by its access to the three companies that account for most of the lighting recycling in Colombia and a 100% participation rate. Interviewers determined that information saturation was reached within each company. Similarities in worker safety, availability of information in the native language, resources, and education level make the findings of this study culturally relevant for recycling of hazard materials elsewhere in South America [120,121,122,123,124,125,126,127].

## 5. Conclusions

This study showed that many operators supplement what they learn in trainings with internet searches, but that what they find online is often not understandable or applicable to them. Therefore, relevant safety materials and research should be made more readily available online. In addition, biomonitoring of e-waste workers’ body burden of mercury (and other work-related chemicals) would provide important information for public health, including comparisons between workers in different roles and at different facilities. Future research could address interventions to reduce mercury exposure, the effectiveness of the machinery used in glass cleaning procedures within the Colombian context, employee retention, and improvements in recycling technology. While in the short-term it is critical to improve health literacy among workers exposed to toxic materials, the duty to prevent exposures falls to the State and businesses to protect their employees, prevent toxic exposures, and promote healthy equity.

## Figures and Tables

**Figure 1 ijerph-18-09295-f001:**
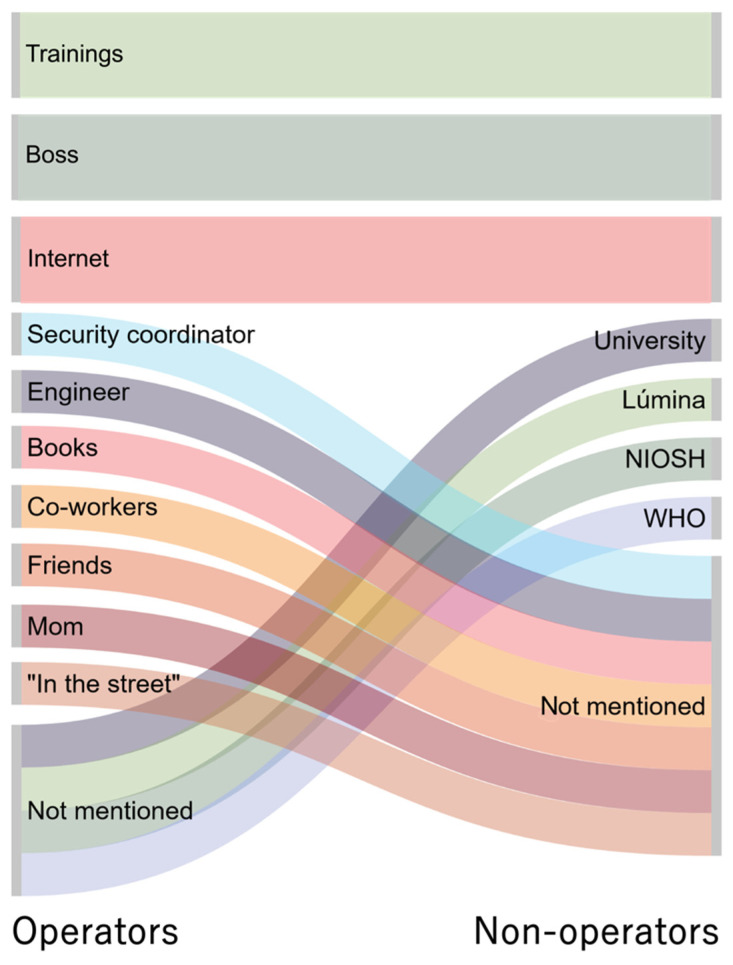
Operators and non-operators used different sources of information about mercury exposure, presented here in a Sankey diagram. Operators’ sources of health risk information are shown on the left; non-operators’ sources are shown on the right. Operators mentioned several informal sources not mentioned by non-operators, while non-operators referred to several additional formal sources of health information not mentioned by operators. Information from training, bosses, and the internet were mentioned most frequently in both groups and are shown in larger boxes. Warm colors indicate informal sources of information (e.g., internet searches, personal contacts) while cool colors indicate formal sources (e.g., on-site trainings, supervisors). Abbreviations: National Institute for Occupational Safety and Health (NIOSH), World Health Organization (WHO).

**Figure 2 ijerph-18-09295-f002:**
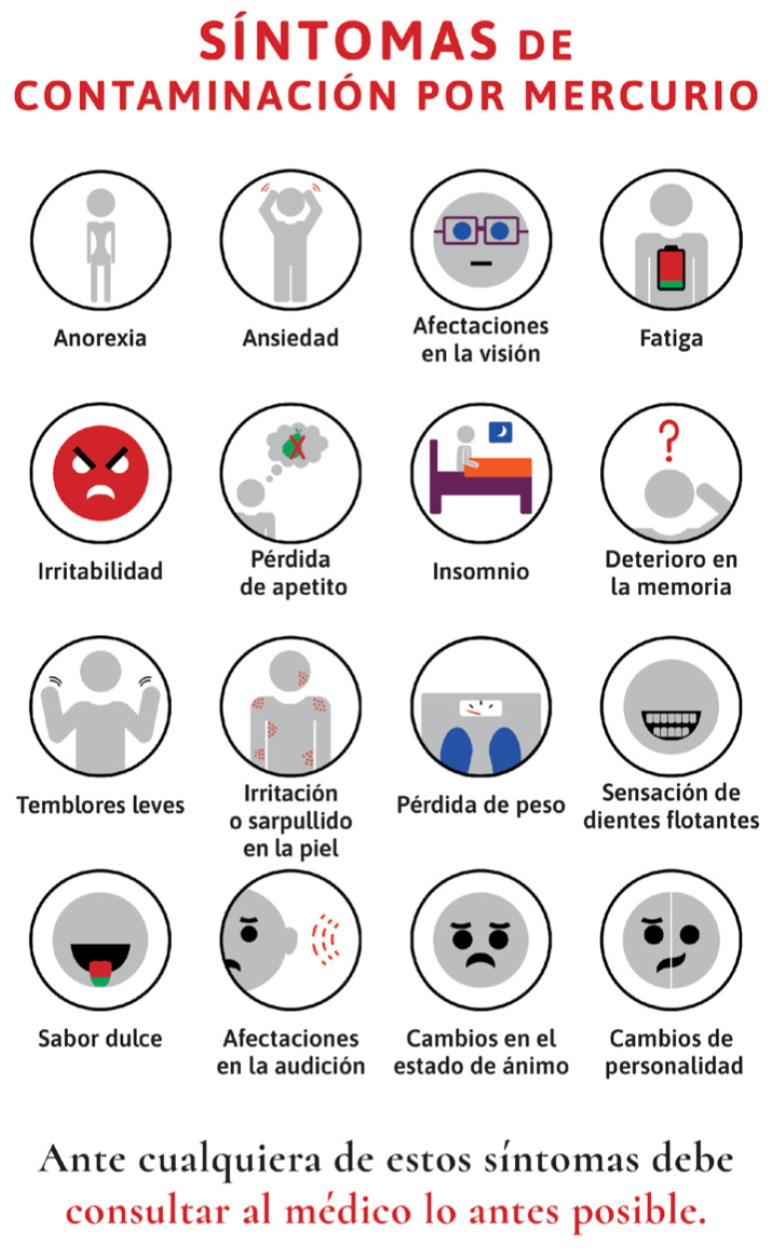
Public health infographics designed based on the findings of this study to communicate body systems affected by mercury vapor exposure. Infographic created by Maria Carolina Cuestas Moncada, Juan David Reyes Estupiñan, and David Felipe Salazar Chiguasuque and reproduced with permission.

**Figure 3 ijerph-18-09295-f003:**
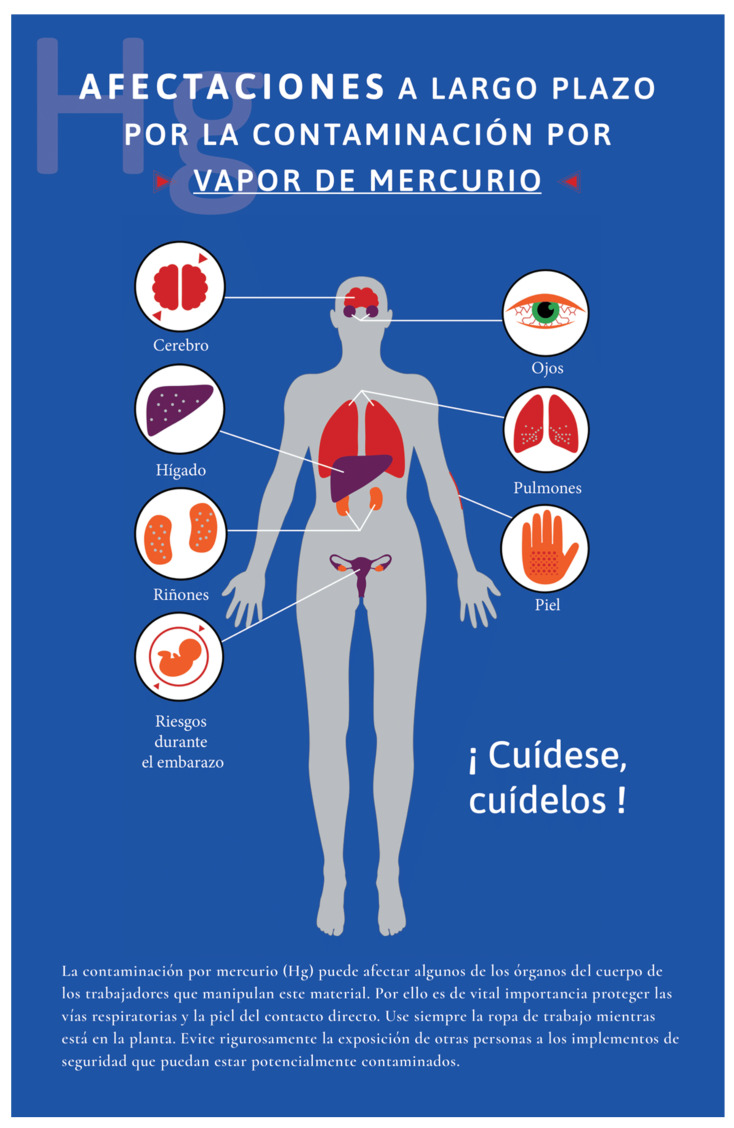
Public health infographics designed based on the findings of this study to visualize potential symptoms related to mercury exposure. Infographic created by Maria Carolina Cuestas Moncada, Juan David Reyes Estupiñan, and David Felipe Salazar Chiguasuque and reproduced with permission.

**Figure 4 ijerph-18-09295-f004:**
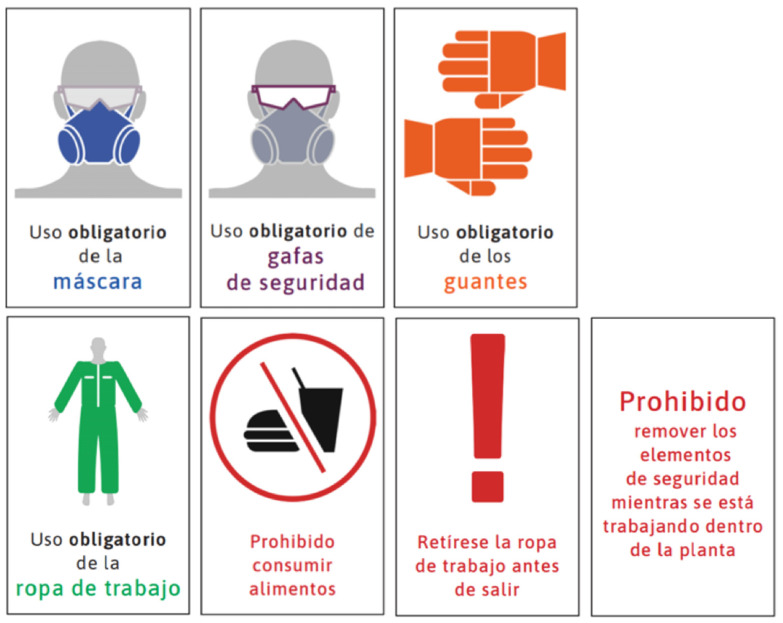
Public health infographics designed based on the findings of this study to communicate the importance of PPE when working with mercury. Infographics highlight (left to right) masks, safety glasses, gloves, work clothes, eating prohibited, removal of work clothes before leaving work, and the prohibited removal of personal protective equipment while working. Infographics created by Maria Carolina Cuestas Moncada, Juan David Reyes Estupiñan, and David Felipe Salazar Chiguasuque and reproduced with permission.

**Table 1 ijerph-18-09295-t001:** Workplace safety improvements recommended by recycling workers to reduce hazardous exposures to mercury.

**Training**	Ensure that all workers get training promptly upon arrival, including videos on mercury Improve organization during training to reduce staff turnover Raise awareness about not drinking water in the warehouse Improve awareness about not taking masks off to “take a breather” Train workers to store filters in a plastic bag to avoid absorbing particles at night More training on how to handle mercury in liquid form
**Facility**	Do not allow the operators to enter the administrative area during their workday; place a filter dispenser within the warehouse for the workers to replace their expired filters without having to contaminate the administrative area while waiting for their supervisor Isolate the bulb processing machine and area where workers handle mercury-containing materials Remove processed glass to make more room to work safely; work with clients to pick up the glass in a timely manner Better organize work areas to avoid accidents Showers at work to clean themselves before going home Place extractors in the area of lighting material recycling Better ventilation in all areas Move the administrative staff to the second floor, as mercury will likely concentrate on the first floor due to its heavy molecular weight [73]. Reduce the need to carry loads up and down the stairs/make each floor self-sufficient so that the workers do not exhaust themselves or overheat going up and down
**Clients**	Work with the clients who provide the bulbs to improve the transportation process and packaging; Ministry of Environment regulations can be used as a guide to avoid breakages that release mercury vapor [74] Improve organization of materials in the transportation process, to avoid cross-contamination from mercury-containing material to e-waste Improve coordination of the arrival of material so the workers are not rushed when multiple trucks arrive at one time Request for manufacturers to put a sticker on each bulb explaining that it contains mercury, and telling the consumer how to properly dispose of it, so that they do not arrive broken
**Tests**	Conduct biomonitoring of workers’ exposure to mercury

**Table 2 ijerph-18-09295-t002:** Interview excerpts from text transcribed in original Spanish.

Interview (Entrevista)	Excerpts (Extractos)
8	“No, no creo que es peligroso.”
1	“No es 100% factor de riesgo.”
14	“Es un poco peligroso para manipular.”
27	“El componente no se elimina, sino que se queda en el ambiente, en el agua, en el organismo, en los alimentos y por eso es un riesgo muy alto.”
6	“Ese se jodió con el mercurio” “hay mucha gente que le tiene miedo a esto.”
12	“Trato no estar cuando hay pico.s”
1	“Hasta el momento es seguro, porque usamos implementos de seguridad”
10	“Igual, usamos la careta y guantes y todo. Todo esta seguro.”
17	“Pero si utilizo los elementos no creo que pueda llegar a afectarme tanto.”
25	“Mmm, pues puede ser si, pero con los elementos que le dan a uno … manejan mucha protección.”
5	“Otras que duran tres días” “Y hay chicos que entran un día, y en medio día se van.”
11	“Porque acá hay una rotación de personal … y los nuevos de pronto no dan importancia a la careta.”
5	“También de la tarea acá … si cartón, plástico, vidrio, pues entonces algunos piensan que es una empresa recicladora.”

## Data Availability

Requests for anonymized data from survey questions and transcripts may be directed to the corresponding authors.

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
