# Peer review of "Mercury Exposure among E-Waste Recycling Workers in Colombia: Perceptions of Safety, Risk, and Access to Health Information"

_ijerph, 2021, doi:10.3390/ijerph18179295_

Round 1

Reviewer 1 Report

The Authors provided a reply to all the questions posed by the Authors.

However, despite the efforts done, the manuscript is still affected by a number of criticisms, as the soundness of the applied methodology is not appropriately demonstrated and data analysis was not clearly explained and appropriately integrated after the revision.

Thus, in the Reviewer's opinion, the manuscript should be rejected.

Reviewer 2 Report

The author has revised this manuscript following our suggestions and has agreed to publish it in its current form.

This manuscript is a resubmission of an earlier submission. The following is a list of the peer review reports and author responses from that submission.

Round 1

Reviewer 1 Report

The manuscript deals with a very interesting topic for the journal. Despite this, the methodology lacks important details, so the data analysis. The Authors are kindly invited to take into account the suggestions in the following.

Please, cross-out the last keyword (in spanish). Check whether the keywords You suggest are in the appropriate maximum number for the Journal.

2.4. Please, specify the number of interviews (out of n=35) was actually used for data analysis (as You said that 10% of the interviews at site 1 was not used due to poor audio quality).

In the caption of Figure 1 the text provide an interpretation of the graph. Please, summarize it according to the typical structure of captions and move the comments to the manuscript text. Of course, this is only a suggestion, but in the Reviewer’s opinion, this will improve clarity.

Line 202: The Reviewer thinks that data need a more in-depth analysis. Just as a comment, You got the 61% of operators perceived a low risk compared to 50% of non-operators. However, 23% of operators perceived a high risk compared to 0% of non-operators, and this needs to be considered.

Table 1. What do the numbers in brackets mean? (namely (122 and 123)

The manuscript lacks a section specifically related to the severity of the risk. All the discussion is based on the perception of risk by the people working in the plant, and this is good. However, no information are provided on the actual risk deriving from the presence of Hg in the plant. Did the Authors performed a risk analysis for the exposure? Are the plants equipped with air extraction and purification (at least in the areas where Hg volatilization may occur?). Are the PPE specifically tailored to Hg (i.e., do they have activated carbon filtration or similar systems?). How PPE are managed and maintained? Are they properly locked at the end of each work shift? How many times they are replaced with new ones?

Another important aspect in the risk estimation is the capacity of the plant (theoretical and actual), how the plant is designed, the presence of air exchange systems, windows and so on. Otherwise, it is not clear whether the score “low”, “medium” or “high” mirrors the actual risks or not.

Table 2. The Reviewer suggests to cross out the Table or to add a column with the English translation. Spanish is the most largely spoken language in the world; however many readers of the Journal can understand English and not Spanish, so that the meaningfulness of the table in the present form is low. If you think that the replies in the original languages add info for the reader, please consider the need of translating them in English.

The data analysis is weak. Please, provide data interpretation in order to gain a more accurate figure of the replies. For example, try to cluster the replies on the basis of: 1) gender; 2) age; 3) level of education; 4) employment age, and so on.

The information strategy “only” refers to symptoms and health effects. This is an important issue, however all the aspects related to the proper utilization and management of PPE, the behaviour to be adopted, the need to verify that safety equipment are properly working and so on, are not mentioned. The Authors are kindly invited to explain the more appropriate ways to reach the targets with this very important messages.

Author Response

1.1.     The manuscript deals with a very interesting topic for the journal. Despite this, the methodology lacks important details, so the data analysis. The Authors are kindly invited to take into account the suggestions in the following.

Response:

The researcher thanks the reviewer for their consideration and careful review of the manuscript.

1.2.     Please, cross-out the last keyword (in spanish). Check whether the keywords You suggest are in the appropriate maximum number for the Journal.

Response: The corresponding changes have been made to the manuscript.

1.3.     2.4. Please, specify the number of interviews (out of n=35) was actually used for data analysis (as You said that 10% of the interviews at site 1 was not used due to poor audio quality).

Response:

We agree that this text could be clearer. 3 interviews had poor audio for approximately 10% of the recorded minutes; however 90% of these 3 interviews was still usable for the analysis. We have now made the following revision (line 177): “All interviews were included in the analysis. Approximately 10% of 3 interviews from company 1 was not able to be transcribed due to poor audio quality; however, no key data is believed to have been lost.”

1.4.     In the caption of Figure 1 the text provide an interpretation of the graph. Please, summarize it according to the typical structure of captions and move the comments to the manuscript text. Of course, this is only a suggestion, but in the Reviewer’s opinion, this will improve clarity.

Response: Thank you. The caption now begins with the following interpretation: “Operators and non-operators used different sources of information about mercury exposure,” and the caption text has been reorganized for clarity.

1.5.     Line 202: The Reviewer thinks that data need a more in-depth analysis. Just as a comment, You got the 61% of operators perceived a low risk compared to 50% of non-operators. However, 23% of operators perceived a high risk compared to 0% of non-operators, and this needs to be considered.

Response:

Thank you for this helpful comment. We have now added the following text as well as additional details to this results section: “Operators also varied more in their perception of their personal exposure than operators, with more operators.” The statement supports the interpretation that operators tend to compare themselves to other workers who they perceive as more exposed than themselves.

1.6.     Table 1. What do the numbers in brackets mean? (namely (122 and 123)

Response: These are citations to references in the reference list.

1.7.     The manuscript lacks a section specifically related to the severity of the risk. All the discussion is based on the perception of risk by the people working in the plant, and this is good. However, no information are provided on the actual risk deriving from the presence of Hg in the plant. Did the Authors performed a risk analysis for the exposure? Are the plants equipped with air extraction and purification (at least in the areas where Hg volatilization may occur?). Are the PPE specifically tailored to Hg (i.e., do they have activated carbon filtration or similar systems?). How PPE are managed and maintained? Are they properly locked at the end of each work shift? How many times they are replaced with new ones?

Response:

Human biomonitoring was not included as part of this study and has been extremely limited in the context of recycling facilities in Colombia. While we would like to study the link between risk perception to actual exposure hazards, this is unfortunately outside the scope of the current study. However we are maintaining a working relationship with relevant partners in the recycling industry and hope to follow up the current study with occupational exposure data.

1.8.     Another important aspect in the risk estimation is the capacity of the plant (theoretical and actual), how the plant is designed, the presence of air exchange systems, windows and so on. Otherwise, it is not clear whether the score “low”, “medium” or “high” mirrors the actual risks or not.

Response: See response to 1.7.

1.9.     Table 2. The Reviewer suggests to cross out the Table or to add a column with the English translation. Spanish is the most largely spoken language in the world; however many readers of the Journal can understand English and not Spanish, so that the meaningfulness of the table in the present form is low. If you think that the replies in the original languages add info for the reader, please consider the need of translating them in English.

Response:

Table 2 is the original Spanish text corresponding to translated quotes used throughout the manuscript text; they were included in Table 2 for transparency. Interview ID numbers used in Table 2 correspond to those used in the manuscript text.

1.10.   The data analysis is weak. Please, provide data interpretation in order to gain a more accurate figure of the replies. For example, try to cluster the replies on the basis of: 1) gender; 2) age; 3) level of education; 4) employment age, and so on.

Response:

Thank you for this feedback. This study is primarily a qualitative study aiming primarily to identify weaknesses in risk perception and gain of health information within this population. Research with this population has been strictly curtailed and the sample population is medium-sized, which makes cluster analysis and analysis of between-group differences difficult to carry out. However, based on the reviewers’ feedback, we have added additional information based on topic frequency during interviews and included new analysis with this revised manuscript.

We have added the following new analyses to the methods and results sections:

Data Analysis (line 184)

“Text was coded for themes and subthemes, assessed for word frequencies, and cross-tabs analyzed for relationships with demographic data, specifically gender, length of employment, age, and self-reported economic strata.”

New results section:

Differences by gender, education, socioeconomic status, and age (line 204)

Results from cross tabulation of demographic data (gender, education, socioeconomic status, and age) with response codes were mostly insignificant, with the exception of a few codes. There were no substantial relationships related to age or education.

Personal protective equipment was brought up 34% more frequently by females than males and general safety measures was brought up 47% more by females than males. Mention of being worried came 40% more from interviews with females than males.

Numerous interviewees reported a lack of awareness within the company to the risks of Hg exposure.  These responses had a relationship with socioeconomic status: 67% of the strata 5 interviewees (highest-income) brought up the issue and 42% of the interviewees from strata 3 – as opposed to 13% in strata 2 and 0% in strata 1.

Transportation and handling of the light bulbs as they relate to workers’ Hg exposure was mentioned by 66% of strata 5 (highest-income) interviewees, and 50% from strata 3. This result is likely related to the fact that employees in management self-reported to be from higher social strata.

Males mentioned specific risky and preferable behaviors happening in the workplace 33% more frequently than females did; however, this was a biased result, as the majority were generated from interviews in Company 1 which was 90% male.

1.11.   The information strategy “only” refers to symptoms and health effects. This is an important issue, however all the aspects related to the proper utilization and management of PPE, the behaviour to be adopted, the need to verify that safety equipment are properly working and so on, are not mentioned. The Authors are kindly invited to explain the more appropriate ways to reach the targets with this very important messages.

Response:

Use of PPE was mentioned frequently in the interviews, and relevant results are presented in section “Response efficacy” (line 292). Based on this reviewer’s feedback, we have incorporated an additional set of infographics as Figure 4 focused on PPE communication, created by the same graphic designers as Figures 2 and 3. These infographics are based on the findings of this study and were created with the intention of being printed as posters and were disseminated to each of the three recycling companies.

Reviewer 2 Report

The article presents an adequate structure, its objective is clear, the results are properly presented, the discussion is in agreement with the results, it presents the limitations as well as the implications for public health.

The main limitation is its exclusive qualitative analysis which limits the causal analysis to being only descriptive and exploratory.

Author Response

2.1.     The article presents an adequate structure, its objective is clear, the results are properly presented, the discussion is in agreement with the results, it presents the limitations as well as the implications for public health.

Response:

The researcher thanks the reviewer for their consideration and careful review of the manuscript.

2.2.     The main limitation is its exclusive qualitative analysis which limits the causal analysis to being only descriptive and exploratory.

Response:

The researcher designed this study as an initial study to explore topics of risk awareness. This study is meant to provide evidence to inform and drive development of health literacy and environmental health awareness campaigns addressing the occupation exposure to Hg. Moreover, e-waste recycling workers’ occupational risks in Latin America have been severely understudied, and it is our aim that this study, while qualitative, may provide motivation and justification for further research - both qualitative and otherwise - by additional research teams.

Reviewer 3 Report

This paper studies how operators and non-operators perceive the risks of their own exposure to mercury. Through interviews with three of the largest formal recycling companies in Colombia (n=35) to learn about these risks, a detailed analysis of the the interview records of non-operating personnel have sufficient interview work, and the analysis of the interview results is also very thorough. However, the interview information comes from the staff’s oral narration and the author’s personal subjective analysis, and lacks a scientific and rigorous evaluation process and specific data results on the degree of harm of mercury to operators and non-operators. We hope to see the data-based impact of mercury on the health of the recyclers, the degree of harm to workers at different working hours and different ages, and the bioavailability of the human body and the degree of specific harm to the human body. So, I recommend that the paper after major revisions, turn to other journals.

Author Response

3.1. This paper studies how operators and non-operators perceive the risks of their own exposure to mercury. Through interviews with three of the largest formal recycling companies in Colombia (n=35) to learn about these risks, a detailed analysis of the interview records of non-operating personnel have sufficient interview work, and the analysis of the interview results is also very thorough.

Response: We thank the reviewer for their time reviewing the manuscript and for this feedback.

3.2 However, the interview information comes from the staff’s oral narration and the author’s personal subjective analysis, and lacks a scientific and rigorous evaluation process and specific data results on the degree of harm of mercury to operators and non-operators. We hope to see the data-based impact of mercury on the health of the recyclers, the degree of harm to workers at different working hours and different ages, and the bioavailability of the human body and the degree of specific harm to the human body. So, I recommend that the paper after major revisions, turn to other journals.

Response:

This study was designed in partnership with stakeholders in Colombia including the Ministry of Environment, The National University of Colombia, and Lumina - a company whose mission is the sustainable management of post-consumer lighting material. Data concerning Hg exposure in workers was being collected in parallel to this study, in a project executed in partnership between Lumina and the Ministry of Environment. This study was meant to complement this biological and environmental data that was gathered. The primary purpose of this study was practical application for planning potential awareness interventions in the lighting material recycling industry.